Functional drug–target–disease network analysis of gene–phenotype connectivity for curcumin in hepatocellular carcinoma

Zhao Yuanyuan 1
Tao Jiahao 2
Chen Zhuangzhong 2
Li Suihui 2
Liu Zeyu 2
Lin Lizhu 2 linlizhu@gzucm.edu.cn
Zhai Linzhu 2 linzhuzhai@163.com
1 State Key Laboratory of Oncology in South China, Sun Yat-sen University Cancer Center , Guangzhou, Guangdong , P. R. China
2 Cancer Center, the First Affiliated Hospital of Guangzhou University of Chinese Medicine , Guangzhou, Guangdong , P. R. China
Pfeffer Ulrich
Electronic publication date: 2021 Oct 26
Publication date: 2021
Volume: 9
Electronic Location ID: e12339
Received 2021 Jun 2; Accepted 2021 Sep 28
Copyright: © 2021 Zhao et al.
Copyright year: 2021
Copyright holder: Zhao et al.
License: This is an open access article distributed under the terms of the Creative Commons Attribution License, which permits unrestricted use, distribution, reproduction and adaptation in any medium and for any purpose provided that it is properly attributed. For attribution, the original author(s), title, publication source (PeerJ) and either DOI or URL of the article must be cited.
License URL: https://creativecommons.org/licenses/by/4.0/

Keywords: Traditional Chinese medicine, Network pharmacology, Cancer pathway, Hepatocellular carcinoma pathway, Curcumin, Chemoprevention

Funding: National Natural Science Foundation of China 81873147 Guangzhou University of Chinese Medicine A1-AFD018171Z11069 The study was supported by grants from the National Natural Science Foundation of China (grant no. 81873147) and the High Level University Project of Guangzhou University of Chinese Medicine (grant no. A1-AFD018171Z11069). The funders had no role in study design, data collection and analysis, decision to publish, or preparation of the manuscript.

==============================
Background

The anti-tumor properties of curcumin have been demonstrated for many types of cancer. However, a systematic functional and biological analysis of its target proteins has yet to be fully documented. The aim of this study was to explore the underlying mechanisms of curcumin and broaden the perspective of targeted therapies.

Methods

Direct protein targets (DPTs) of curcumin were searched in the DrugBank database. Using the STRING database, the interactions between curcumin and DPTs and indirect protein targets (IPTs) weres documented. The protein–protein interaction (PPI) network of curcumin-mediated proteins was visualized using Cytoscape. Kyoto Encyclopedia of Genes and Genomes (KEGG) pathway enrichment analysis was performed for all curcumin-mediated proteins. Furthermore, the cancer targets were searched in the Comparative Toxicogenomics Database (CTD). The overlapping targets were studied using Kaplan–Meier analysis to evaluate cancer survival. Further genomic analysis of overlapping genes was conducted using the cBioPortal database. Lastly, MTT, quantitative polymerase chain reaction (qPCR), and western blot (WB) analysis were used to validate the predicted results on hepatocellular carcinoma (HCC) cells.

Results

A total of five DPTs and 199 IPTs were found. These protein targets were found in 121 molecular pathways analyzed via KEGG enrichment. Based on the anti-tumor properties of curcumin, two pathways were selected, including pathways in cancer (36 genes) and HCC (22 genes). Overlapping with 505 HCC-related gene sets identified in CTD, five genes (TP53, RB1, TGFB1, GSTP1, and GSTM1) were finally identified. High mRNA levels of TP53, RB1, and GSTM1 indicated a prolonged overall survival (OS) in HCC, whereas elevated mRNA levels of TGFB1 were correlated with poor prognosis. The viability of both HepG2 cells and Hep3B cells was significantly reduced by curcumin at concentrations of 20 or 30 μM after 48 or 72 h of culture. At a concentration of 20 μM curcumin cultured for 48 h, the expression of TGFB1 and GSTP1 in Hep3B cells was reduced significantly in qPCR analysis, and reduced TGFB1 protein expression was also found in Hep3B cells.

Introduction

Primary liver cancer is the sixth most common cancer and the second largest cause of cancer mortality in the world (Ferlay et al., 2015). Hepatocellular carcinoma (HCC) constitutes approximately 80% of all primary liver cancer (McGlynn, Petrick & London, 2015). The main causes of HCC are hepatitis B and C virus infection, alcoholic liver disease, non-alcoholic fatty liver diseases related to obesity and/or diabetes, aflatoxin B1 contamination of food, and less frequently, genetic predisposition. The incidence of HCC is increasing worldwide, mostly due to the spread of chronic infections with hepatitis B and C viruses. Most of these risk factors cause persistent inflammation in the liver, leading to the formation of liver cirrhosis and then HCC.

Curcumin, also known as diferuloylmethane, is the active ingredient of the dietary spice occurring in the rhizomes of Curcuma longa, a plant belonging to the ginger family (Fig. 2A). Extensive studies investigating curcumin over the past few decades revealed the health benefits of curcumin, including anticancer, anti-inflammatory, antioxidant, and hypoglycemic effects (Prasad et al., 2014). In recent years, curcumin has been increasingly recognized for its anti-tumor and chemopreventive properties, especially in gastrointestinal tumors (Huminiecki, Horbanczuk & Atanasov, 2017). For example, curcumin has shown chemopreventive effects in animal models of colon cancer (Kawamori et al., 1999), stomach cancer (Huang et al., 1994), and HCC (Chuang et al., 2000). Curcumin has been reported to decrease cell growth and induce apoptosis mainly through the inhibition of nuclear factor kappa-B (NFκB) (Ghasemi et al., 2019). Recent reports also indicate the curcumin could decrease tumor cell viability by inhibiting the activity of ATP synthase in isolated mitochondrial membranes which leads to a dramatic drop of ATP and a reduction of oxygen consumption (Bianchi et al., 2018). In addition, DYRK2 has been identified as a direct target of curcumin. By diminishes DYRK2-mediated 26S proteasome phosphorylation in cells, curcumin leading to reduced proteasome activity and inhibited cell proliferation (Banerjee et al., 2018). Several phase I and II clinical trials with curcumin have been conducted for the treatment of different types of cancer (Gupta, Patchva & Aggarwal, 2013; Hatcher et al., 2008).

Figure 1 Workflow chart of study.

Figure 2 (A) The structure of curcumin. (B) The interactions of 5 DPTs of curcumin by STRING with a minimum required interaction score 0.2. (C) PPI network of curcumin-mediated proteins analyzed by Cytoscape. Network nodes represent proteins, and edges represent protein-protein associations. High node degree values are represented by big sizes. (D) The top 15 statistically enriched KEGG pathways and involved gene numbers. (E) Venn diagrams for intersections of three gene sets. (F) A visual display of the network connected to 5 selected protein targets (TP53, RB1, TGFB1, GSTP1, and GSTM1). Neighboring proteins connected to the five query proteins, filtered from no more than five interactors to no interactor. (G) Overview of genetic alterations related to curcumin-associated genes in genomics data sets available in seven different HCC studies in cBioPortal databases.

Recent studies using data mining techniques for the analysis of specific bioinformatics domains have made great progress (Lan et al., 2018). Accordingly, a strategy to systematically explore the rich data publicly available and the underlying connectivity between gene and phenotype mediated by curcumin should be helpful. With advances in genomics, data network analysis has been used to effectively analyze candidate genes linked to experimentally verifiable pathways via mining of web-accessible, open portal databases (Hsieh et al., 2016). In addition, an in-depth analysis of the databases may uncover, previously unknown relationships between drugs, targets, and cancers. These results can be used to generate authentic and rational leads for further investigations. Recently, the combination of related databases including DrugBank, STRING, cBio Cancer Genomics Portal (cBioPortal), and Comparative Toxicogenomics Database (CTD) has been utilized for drug–target–cancer network analysis (Fu et al., 2019; Zhang et al., 2018). In this study, DrugBank was employed to broadly analyze curcumin and drug-target data to obtain related direct protein targets (DPTs). Interactions between DPTs and indirect protein targets (IPTs) were predicted with the STRING database. The readout DPTs and IPTs were further analyzed for functionality via Kyoto Encyclopedia of Genes and Genomes (KEGG) pathway enrichment analysis with STRING. The pivotal genes were obtained via intersections of the CTD and STRING databases, and the genomic alterations were investigated using the cBioPortal database. Our findings provide information to obtain a better understanding of the anti-tumor mechanisms of curcumin and identify new targets in the treatment of HCC by curcumin.

Materials and Methods

Search for DPTs of curcumin

The DrugBank database (https://www.drugbank.ca) is a richly annotated resource that combines detailed drug data with comprehensive drug target and drug action information (Wishart et al., 2008). In this study, the latest release of DrugBank database (version 5.1.4) (Wishart et al., 2018) was employed to search for the interactions between curcumin and its DPTs and generate a curcumin-target network. The word “Curcumin” was searched as a keyword under drug classification entry.

PPI network generation

The STRING database (http://www.string-db.org/) provides a critical assessment and integration of protein-protein interactions, including direct (physical) as well as indirect (functional) associations (Szklarczyk et al., 2015). Using the latest STRING database (version 11.0) (Szklarczyk et al., 2019) search function, data underlying the interaction between DPTs and IPTs were first generated for curcumin by setting a minimum required interaction score of 0.5 and a maximum number of interactors of 50. The data were integrated into a curcumin-mediated network and visualized using Cytoscape (version 3.7.1) (Su et al., 2014), which is an open-source software platform for the visualization of complex networks and their integration with any attribute (Shannon et al., 2003).

KEGG pathway enrichment analysis and overlapping HCC genes

Biochemical pathways linked to curcumin DPT/IPT interactions were navigated via the KEGG pathway enrichment analysis tool in the STRING database. The top 15 pathways with a False Discovery Rate (FDR) of less than 1.00E−10 were selected. To identify disease targets for HCC, “Hepatocellular carcinoma” was searched for in CTD (http://www.ctdbase.org/). CTD is a robust, publicly available database that aims to advance the understanding of environmental exposures and their impacts on human health. It provides manually curated information about chemical–gene/protein interactions, and chemical–disease and gene-disease relationships (Davis et al., 2019). Overlapping genes of HCC and cancer-related pathways and HCC targets were selected for further analysis.

Exploring cancer genomics data linked to curcumin

The cBioPortal (https://www.cbioportal.org/) for Cancer Genomics provides a web resource for exploration, visualization, and analysis of multidimensional cancer genomics data (Cerami et al., 2012; Gao et al., 2013). In this study, the screened genes from the foregoing investigation were assessed in all HCC studies available in cBioPortal databases. Using the portal search function, curcumin-related genes in HCC were evaluated for genomic alteration, network analysis was performed, and mutual exclusivity or concurrent relationships between gene pairs of the same gene set were identified. The study with the largest sample size was chosen to analyze the screened proteins in cBioPortal databases to further interpret the results.

Analysis of curcumin-associated genomics datasets and HCC survival

The Kaplan–Meier method (http://kmplot.com/) (Nagy et al., 2018) was used to evaluate the overall survival (OS) in 364 HCC samples with altered genes. OS was defined as the time to death. A two-tailed P value of less than 0.05 was considered statistically significant.

Cell culture

HepG2 and Hep3B cells were purchased from Cell Cook (http://www.cellcook.com/). They were authenticated using short tandem repeat matching analysis and cultured in DMEM supplemented with 10% v/v FBS and 1% v/v penicillin/streptomycin in a CO2 incubator at 37 °C and 95% relative humidity.

Cell viability test

The MTT (3-(4,5)-dimethylthiahiazo (-z-y1)-3,5-di-phenytetrazoliumromide) assay was used to assess cell viability. First, 100 μL of cells (about 5,000–10,000 cells) was transfused into a 96-well culture plate, which was then placed in a hatching house overnight for pre-culture (37 °C, 5% CO2). According to the drug concentration, all cells were divided into five groups. After pre-culture, 10 μL of drug solution was added to every well (the drug concentrations were 0, 5, 10, 20, 30 μM) and incubated at 37 °C. After that, 10 μL MTT solution was added to each well at 24, 48, and 72 h and incubated at 37 °C for 4 h. The absorbance value of each hole at 450 nm was determined, and the cell vitality value was calculated. The cell vitality formula is as follows: Cell viability% = (OD in experimental group – OD in blank group)/(OD in control group – OD in blank group) × 100%.

Extraction of total RNA and reverse transcription-quantitative PCR (RT-qPCR)

Cells of diffetent group were collected after being treated and cultured, and total RNA was extracted using TRIzol® reagent. Oligo (dT) primers and M-MLV reverse transcriptase were used to synthesize cDNA under the instruction of protocol. PCR amplification was performed on an RT-qPCR instrument with a GoTaq® qPCR Master mix. The relative gene expression of each sample was determined with the 2-ΔΔCq method.

Western blot analysis

Protein extractions were prepared with a modified RIPA buffer (Beyotime Institute of Biotechnology, Haimen, Jiangsu, China) with 0.5% SDS. Proteinase inhibitor cocktail (Beyotime Institute of Biotechnology, Haimen, Jiangsu, China) were involved in the whole process. Protein samples was separated using SDSPAGE, and then transferred to a PVDF membrane. The membranes were blocked with 5% BSA in TBS (containing 0.05% Tween-20) and incubated overnight with the primary antibodies. After the membranes were washed, the membranes were incubated with horseradish peroxidase-linked immunoglobulin G secondary antibody (Ms., Jackson ImmunoResearch) for 2 h. The membranes were developed using an enhanced chemiluminescence system (Thermo Scientific Pierce, Waltham, MA, USA).

Results

Characterization of curcumin DPTs

A flow chart of study steps was present in Fig. 1. Drugbank was queried using curcumin as the keyword to identify the bioactivities and to determine the DPTs of curcumin and retrieve relevant information. The result showed an accession number of DB11672 and classified curcumin as a highly pleiotropic molecule with anti-tumor, antibacterial, anti-inflammatory, hypoglycemic, antioxidant, wound-healing, and antimicrobial activities (Gupta, Patchva & Aggarwal, 2013). Clinical data showed that curcumin is undergoing clinical trials for colon cancer (NCT02724202), prostate cancer (NCT03769766), breast cancer (NCT03980509), and lung cancer (NCT02321293). Subsequent screening demonstrated five DPTs in human beings. Table 1 summarizes the five DPTs of curcumin: PPARG, VDR, ABCC5, CBR1, and GSTP1. In addition, the interactions between five DPTs were analyzed by STRING and illustrated in Fig. 2B.

Table 1 Direct protein targets of curcumin identified using DrugBank.

Uniprot ID	Gene name	Targets	General function	
P37231	PPARG	Peroxisome proliferator-activated receptor gamma	Zinc ion binding	
P11473	VDR	Vitamin D3 receptor	Zinc ion binding	
O15440	ABCC5	Multidrug resistance-associated protein 5	Organic anion transmembrane transporter activity	
P16152	CBR1	Carbonyl reductase [NADPH] 1	Prostaglandin-e2 9-reductase activity	
P09211	GSTP1	Glutathione S-transferase P	S-nitrosoglutathione binding	

Characterization of curcumin IPTs and visualization of PPI network construction

Expanding the search using STRING database, a total of 204 target proteins of curcumin were detected, including 199 IPTs, which were related to five DPTs (Table S1). The dataset obtained was integrated to construct a biological network using Cytoscape. As shown in Fig. 2C, 1962 PPI pairs were found in the network of curcumin-mediated proteins (Table S2). In this network, nodes represent proteins, and edges denote protein-protein associations. In addition, the node degree, which indicates the centrality of proteins, was calculated using CentiScaPe 2.2 (Table S3). As a result, 14 proteins including four DPT targets (GSTP1, VDR, CBR1, and PPARG) with a degree value ≥ 50 are shown in Table 2 and displayed in Fig. 2C with larger node sizes.

Table 2 Protein targets with degree value ≥ 50.

Gene	Node degree value	Gene	Node degree value	
EP300	75	NCOA1	52	
CREBBP	72	CYP3A4	51	
RXRA	68	GSTP1	50	
NCOR1	60	NCOA3	50	
CYP2E1	58	VDR	50	
CYP1A1	54	CBR1	50	
NCOR2	53	PPARG	50	

KEGG enrichment pathway analysis of proteins connected to curcumin

To identify the functional features of curcumin-mediated targets, KEGG pathway enrichment analysis was performed using STRING. Target genes were found in 121 molecular pathways in KEGG enrichment (Table S4). As shown in Table 3 and Fig. 2D, the top 15 KEGG pathways connected to curcumin-mediated proteins included metabolism of xenobiotics by cytochrome P450 (52 genes), chemical carcinogenesis (50 genes), drug metabolism-cytochrome P450 (44 genes), retinol metabolism (32 genes), steroid hormone biosynthesis (30 genes), drug metabolism-other enzymes (31 genes), pentose and glucuronate interconversion (22 genes), ascorbate and aldarate metabolism (20 genes), metabolic pathways (65 genes), glutathione metabolism (20 genes), porphyrin and chlorophyll metabolism (19 genes), arachidonic acid metabolism (18 genes), pathways in cancer (36 genes), HCC (22 genes), and fluid shear stress and atherosclerosis (19 genes). These KEGG enrichment pathways showed functional features of curcumin gene sets and indicated that curcumin-mediated proteins were mainly associated with basal metabolism and cancer-related pathways. Based on the anti-tumor properties of curcumin reported in diverse malignant tumors, two pathways were selected, including pathways in cancer (36 genes) and HCC (22 genes). This result suggested that HCC might be used as a phenotype connected to curcumin-mediated proteins. In addition, to further identify disease targets for HCC, the phrase “Carcinoma, Hepatocellular” was searched in CTD, and a total of 505 genes with either a curated association or an inferred association via a curated chemical interaction with the disease were identified (Table S5). These genes were marked with “T,” which indicates a gene that is or may be a therapeutic disease target, or “M,” which denotes a gene that may be a disease biomarker or involved in the etiology of a disease. Five overlapping genes (TP53, RB1, TGFB1, GSTP1, and GSTM1) resulting from the intersections between pathways in cancer (36 genes), HCC pathway (22 genes), and targets for HCC by CTD (505 genes) were visualized using Venn diagrams and STRING (Figs. 2E and 2F).

Table 3 Top 15 enriched KEGG pathways identified using STRING.

Pathways	Gene count	FDR	Matching proteins	
Metabolism of xenobiotics by cytochrome P450	52	5.68E−70	ADH1A,ADH1B,ADH1C,ADH5,AKR1C1,AKR7A2,ALDH3A1,CBR1,CBR3,CYP1A1,CYP1A2,CYP1B1,CYP2A13,CYP2A6,CYP2B6,CYP2C9,CYP2D6,CYP2E1,CYP2F1,CYP3A4,CYP3A5,ENSG00000270386,EPHX1,GSTA1,GSTM1,GSTO1,GSTO2,GSTP1,GSTT2B,HPGDS,HSD11B1,MGST1,MGST2,MGST3,UGT1A1,UGT1A10,UGT1A3,UGT1A4,UGT1A5,UGT1A6,UGT1A7,UGT1A8,UGT1A9,UGT2A2,UGT2A3,UGT2B10,UGT2B11,UGT2B15,UGT2B17,UGT2B28,UGT2B4,UGT2B7	
Chemical carcinogenesis	50	1.80E−65	ADH1A,ADH1B,ADH1C,ADH5,AKR1C2,ALDH3A1,CBR1,CYP1A1,CYP1A2,CYP1B1,CYP2A13,CYP2A6,CYP2C18,CYP2C19,CYP2C8,CYP2C9,CYP2E1,CYP3A4,CYP3A5,ENSG00000270386,EPHX1,GSTA1,GSTM1,GSTO1,GSTO2,GSTP1,GSTT2B,HPGDS,HSD11B1,MGST1,MGST2,MGST3,UGT1A1,UGT1A10,UGT1A3,UGT1A4,UGT1A5,UGT1A6,UGT1A7,UGT1A8,UGT1A9,UGT2A2,UGT2A3,UGT2B10,UGT2B11,UGT2B15,UGT2B17,UGT2B28,UGT2B4,UGT2B7	
Drug metabolism-cytochrome P450	44	1.30E−57	ADH1A,ADH1B,ADH1C,ADH5,ALDH3A1,CYP1A2,CYP2A6,CYP2B6,CYP2C19,CYP2C8,CYP2C9,CYP2D6,CYP2E1,CYP3A4,CYP3A5,ENSG00000270386,GSTA1,GSTM1,GSTO1,GSTO2,GSTP1,GSTT2B,HPGDS,MGST1,MGST2,MGST3,UGT1A1,UGT1A10,UGT1A3,UGT1A4,UGT1A5,UGT1A6,UGT1A7,UGT1A8,UGT1A9,UGT2A2,UGT2A3,UGT2B10,UGT2B11,UGT2B15,UGT2B17,UGT2B28,UGT2B4,UGT2B7	
Retinol metabolism	32	8.99E−39	ADH1A,ADH1B,ADH1C,ADH5,CYP1A1,CYP1A2,CYP2A6,CYP2B6,CYP2C18,CYP2C8,CYP2C9,CYP3A4,CYP3A5,ENSG00000270386,UGT1A1,UGT1A10,UGT1A3,UGT1A4,UGT1A5,UGT1A6,UGT1A7,UGT1A8,UGT1A9,UGT2A2,UGT2A3,UGT2B10,UGT2B11,UGT2B15,UGT2B17,UGT2B28,UGT2B4,UGT2B7	
Steroid hormone biosynthesis	30	2.17E−36	AKR1C1,AKR1C2,AKR1C3,AKR1C4,CYP1A1,CYP1A2,CYP1B1,CYP2E1,CYP3A4,CYP3A5,ENSG00000270386,HSD11B1,UGT1A1,UGT1A10,UGT1A3,UGT1A4,UGT1A5,UGT1A6,UGT1A7,UGT1A8,UGT1A9,UGT2A2,UGT2A3,UGT2B10,UGT2B11,UGT2B15,UGT2B17,UGT2B28,UGT2B4,UGT2B7	
Drug metabolism-other enzymes	31	3.65E−35	CYP2A6,CYP2E1,CYP3A4,ENSG00000270386,GSTA1,GSTM1,GSTO1,GSTO2,GSTP1,GSTT2B,MGST1,MGST2,MGST3,UGT1A1,UGT1A10,UGT1A3,UGT1A4,UGT1A5,UGT1A6,UGT1A7,UGT1A8,UGT1A9,UGT2A2,UGT2A3,UGT2B10,UGT2B11,UGT2B15,UGT2B17,UGT2B28,UGT2B4,UGT2B7	
Pentose and glucuronate interconversions	22	3.05E–28	AKR1A1,AKR1B1,ENSG00000270386,KL,UGT1A1,UGT1A10,UGT1A3,UGT1A4,UGT1A5,UGT1A6,UGT1A7,UGT1A8,UGT1A9,UGT2A2,UGT2A3,UGT2B10,UGT2B11,UGT2B15,UGT2B17,UGT2B28,UGT2B4,UGT2B7	
Ascorbate and aldarate metabolism	20	1.50E–26	ALDH9A1,ENSG00000270386,UGT1A1,UGT1A10,UGT1A3,UGT1A4,UGT1A5,UGT1A6,UGT1A7,UGT1A8,UGT1A9,UGT2A2,UGT2A3,UGT2B10,UGT2B11,UGT2B15,UGT2B17,UGT2B28,UGT2B4,UGT2B7	
Metabolic pathways	65	3.44E–26	ADH1A,ADH1B,ADH1C,ADH5,ADPGK,AKR1A1,AKR1B1,AKR1C3,AKR1C4,ALDH3A1,ALDH9A1,CBR1,CBR3,CYP1A1,CYP1A2,CYP24A1,CYP27A1,CYP27B1,CYP2A6,CYP2B6,CYP2C18,CYP2C19,CYP2C8,CYP2C9,CYP2E1,CYP2R1,CYP3A4,CYP3A5,DHFR,DHFRL1,ENSG00000270386,GCH1,GCLC,GGT1,GSS,GSTZ1,HPGDS,HSD11B1,KL,MCEE,PCK1,PRDX6,PTGES,PTGES2,PTGES3,PTS,SPR,UGT1A1,UGT1A10,UGT1A3,UGT1A4,UGT1A5,UGT1A6,UGT1A7,UGT1A8,UGT1A9,UGT2A2,UGT2A3,UGT2B10,UGT2B11,UGT2B15,UGT2B17,UGT2B28,UGT2B4,UGT2B7	
Glutathione metabolism	20	1.62E–22	GCLC,GGT1,GPX1,GPX2,GPX3,GPX4,GPX7,GPX8,GSR,GSS,GSTA1,GSTM1,GSTO1,GSTO2,GSTP1,GSTT2B,HPGDS,MGST1,MGST2,MGST3	
Porphyrin and chlorophyll metabolism	19	3.09E–22	ENSG00000270386,UGT1A1,UGT1A10,UGT1A3,UGT1A4,UGT1A5,UGT1A6,UGT1A7,UGT1A8,UGT1A9,UGT2A2,UGT2A3,UGT2B10,UGT2B11,UGT2B15,UGT2B17,UGT2B28,UGT2B4,UGT2B7	
Arachidonic acid metabolism	18	2.35E–18	AKR1C3,CBR1,CBR3,CYP2B6,CYP2C19,CYP2C8,CYP2C9,CYP2E1,GGT1,GPX1,GPX2,GPX3,GPX7,GPX8,HPGDS,PTGES,PTGES2,PTGES3	
Pathways in cancer	36	1.05E–17	CEBPA,CREBBP,EP300,FGF23,FOS,GSTA1,GSTM1,GSTO1,GSTO2,GSTP1,GSTT2B,HDAC1,JUN,MAPK8,MGST1,MGST2,MGST3,NCOA1,NCOA3,NFKB1,NQO1,PAX8,PPARG,RARA,RARB,RASSF1,RB1,RELA,RXRA,RXRB,RXRG,SMAD3,SMAD4,TGFB1,TP53,TRAF2	
HCC	22	4.14E–16	ACTL6A,ARID1A,GSTA1,GSTM1,GSTO1,GSTO2,GSTP1,GSTT2B,MGST1,MGST2,MGST3,NQO1,RB1,SMAD3,SMAD4,SMARCA4,SMARCC1,SMARCC2,SMARCD1,SMARCE1,TGFB1,TP53	
Fluid shear stress andatherosclerosis	19	2.45E–14	FOS,GSTA1,GSTM1,GSTO1,GSTO2,GSTP1,GSTT2B,JUN,MAPK8,MGST1,MGST2,MGST3,NFKB1,NQO1,PIAS4,RELA,SUMO2,TNF,TP53	
Note:

FDR, false discovery rate.

Genetic alterations connected with curcumin-associated genes in HCC

To further validate the link between curcumin-associated genes and HCC, cBioPortal databases were used to explore the five genes (TP53, RB1, TGFB1, GSTP1, and GSTM1) associated with curcumin in HCC. Seven HCC studies were included in cBioPortal (Ahn et al., 2014; Fujimoto et al., 2012; Harding et al., 2019; Pilati et al., 2014; Schulze et al., 2015; Zheng et al., 2018), and the five selected overlapping genes were queried. The results showed that 363 (33%) of the 1,135 samples in seven studies had alterations in one or more of these genes. Alterations ranged from 0.2% to 29% for gene sets submitted for analysis (Fig. 3A). The five genes (TP53, RB1, TGFB1, GSTP1 and GSTM1) carried five gene pairs showing mutual exclusivity alterations, and another five gene pairs showed concurrent alterations (with TP53 and RB1 showing a statistically significant alteration, Table 4). Furthermore, a liver HCC study (TCGA, Provisional), which carried the largest sample size, was selected to analyze the screened proteins in cBioPortal databases. The results showed that 156 (35%) patients/samples carried an alteration in at least one of the five genes queried using OncoPrint; the frequency of alteration in each gene is shown in Fig. 2G. Most gene alterations included mutations and deletions. Most of the TP53 alterations were classified into missense mutations, truncating mutations, and deep deletions. Gene changes associated with RB1 included truncating mutations and deep deletions, whereas for TGFB1, GSTP1 and GSTM1 amplifications were the most common gene alterations.

Figure 3 (A) A visual heatmap of mRNA-level alterations based on five genes (TP53, RB1, TGFB1, GSTP1, and GSTM1) across a HCC study (data taken from the Liver HCC (TCGA, Provisional) study) in cBioPortal databases. Each row represents a gene, and each column represents a tumor sample.Survival analysis of five selected genes according to mRNA expression in HCC: (B) TP53, (C) RB1, (D) GSTM1, (E) GSTP1, and (F) TGFB1. Magenta lines indicate high levels of mRNA expression, while black lines indicate low levels of mRNA expression.

Table 4 Mutual exclusivity analysis of five genes (TP53, RB1, TGFB1, GSTP1, and GSTM1) in seven studies associated with HCC.

A	B	Log 2 odds ratio	p-Value	Tendency	
TP53	RB1	0.755	0.024	Co-occurrence	
RB1	TGFB1	2.131	0.108	Co-occurrence	
TP53	GSTP1	1.045	0.187	Co-occurrence	
TP53	TGFB1	0.564	0.421	Co-occurrence	
RB1	GSTP1	<−3	0.437	Mutual exclusivity	
TP53	GSTM1	1.299	0.495	Co-occurrence	
RB1	GSTM1	<−3	0.861	Mutual exclusivity	
TGFB1	GSTP1	<−3	0.918	Mutual exclusivity	
GSTP1	GSTM1	<−3	0.979	Mutual exclusivity	
TGFB1	GSTM1	<−3	0.985	Mutual exclusivity	

Curcumin-associated genes and survival in HCC

Kaplan–Meier analysis was used to perform survival analysis of patients with HCC based on the five selected genes (TP53, RB1, TGFB1, GSTP1 and GSTM1). A total of 364 patients were involved in this analysis for OS (Menyhárt, Nagy & Győrffy, 2018). As shown in Figs. 3B–3F, high mRNA levels of TP53, RB1, and GSTM1 indicated increased OS in HCC, whereas elevated mRNA levels of TGFB1 were correlated with poor prognosis in the same group of patients.

Curcumin-induced decrease in the activity of HepG2 and Hep3B cells

The test results showed that cell viability appeared dependent on the concentration of curcumin. For Hep3B cells, curcumin (10 μM) significantly decreased cell viability after 48 and 72 h, and 20 and 30 μM of curcumin were found to significantly reduce cell viability after 24, 48, and 72 h (Fig. 4A). For HepG2 cells, curcumin (20 and 30 μM) significantly decreased the cell viability after 48 and 72 h (Fig. 4B). According to these results, the concentration of curcumin was chosen to be 20 μM and the duration of treatment was chosen to be 48 h for subsequent testing.

Figure 4 Validation experiment in vitro (A) Cell viability inhibition experiment of HepG2. (B) Cell viability inhibition experiment of Hep3B. (C) qPCR analysis of five overlapping genes in HepG2. (D) qPCR analysis of five overlapping genes in Hep3B. (E) Western blot analysis of TGFB1 in HepG2. (F) Quantitative analysis of TGFB1 protein expression in HepG2. (G) Western blot analysis of TGFB1 in Hep3B. (H) Quantitative analysis of TGFB1 protein expression in Hep3B.

Curcumin regulation of TGFB1 and GSTP1 expression in Hep3B cells

The expression of TP53, RB1, TGFB1, GSTP1, and GSTM1 was assessed using qPCR in Hep3B cells treated with 20 μM curcumin for 48 h. The results showed that 20 μM curcumin regulated the expression of TGFB1 and GSTP1 in Hep3B cells, but it had no significant effect on the expression of other genes (Figs. 4C and 4D). After treatment of Hep3B cells with 0 and 20 μM curcumin for 48 h, the levels of TGFB1 and GSTP1 protein were detected via western blot. Because the protein background expression of GSTP1 was low, the expression of GSTP1 at the protein level failed to be detected. The results showed that 20 μM curcumin did not regulate the expression of TGFB1 in HepG2 cells (Figs. 4E, 4F), but moderately inhibited thfie expression of TGFB1 in Hep3B cells (Figs. 4G, 4H).

Discussion

Over the past several years, numerous studies have evaluated the effects of curcumin and its analogs in diverse cancers in vitro and in vivo. According to these studies, the potential use of curcumin as a chemopreventive and therapeutic agent in cancers depends on its potent antioxidant and anti-inflammatory activities as well as its ability to modulate various molecular signaling mechanisms (Gupta, Patchva & Aggarwal, 2013; Hatcher et al., 2008; Huminiecki, Horbanczuk & Atanasov, 2017; Prasad et al., 2014). Nevertheless, the mechanism underlying the wide range of anti-cancer effects of curcumin remains incomplete. Current knowledge about curcumin’s functions and mechanisms is based on conventional experiments, which have yet to be fully integrated and understood. Therefore, new analytical methods are needed to correlate curcumin with its target proteins and the observed biological effects. Functional/activity network (FAN) (Hsieh et al., 2016) is a new analytical method, which elucidates the molecular mechanisms of a drug and its association with clinical outcomes in cancer by using a set of web-based tools, such as DrugBank, STRING, cBioPortal, CTD, or Cytoscape. Using this method, a functional drug–target–cancer network analysis of gene–phenotype connectivity associated with curcumin was conducted in this study. This study bridged curcumin with its primary or secondary targets and illustrated the underlying mechanisms of curcumin and its clinical outcomes in HCC.

The effects of curcumin and its analogs have been a subject of investigation over the past decade in preclinical models of HCC (Darvesh, Aggarwal & Bishayee, 2012), but research in this field is far from complete. A search of PubMed using “curcumin” as the keyword returned more than 13,000 publications, whereas the phrase “curcumin and hepatocellular carcinoma” returned only 192 publications. By using FAN, more direct mechanisms may be mined with reasonable experimental feasibility to validate hypotheses explaining the effects of curcumin in HCC.

In this study, the feasibility of FAN analysis for identifying the underlying connectivity between curcumin and HCC was demonstrated. A network including five genes (TP53, RB1, TGFB1, GSTP1, and GSTM1) as targets of curcumin in HCC was identified via a functional drug–target–cancer network analysis. Among the five genes, TP53 and RB1 are tumor suppressor genes, which is consistent with the finding in this study that their high transcriptional expression is correlated with better OS in HCC. Similar to other cancers, TP53 and RB1 are the most commonly inactivated or mutated genes in the case of HCC. Moreover, mutual exclusivity analysis revealed a tendency toward concurrence between TP53 and RB1. Previous studies indicated that curcumin may inhibit cancer growth and induce apoptosis in colon cancer cells (Dasiram et al., 2017), ameliorate the in vitro efficacy of carfilzomib in human multiple myeloma cells (Allegra et al., 2018), promote apoptosis in non-small cell lung cancer (Ye et al., 2015), and inhibit cell growth in nasopharyngeal carcinoma mediated via the TP53 signaling pathway (Wu et al., 2014). Debata et al. (2013) reported that the sunitinib-curcumin combination was effective in restoring the tumor suppressor activity of the RB gene in renal cancer cells. Furthermore, Su, Wang & Chiu (2010) reported that curcumin significantly enhanced p53 or markedly inhibited the RB pathway by suppressing RB phosphorylation in the signaling pathways of glioblastoma (Su, Wang & Chiu, 2010). These studies demonstrated the anti-tumor activity of curcumin and prompted further investigation of curcumin in HCC.

In the case of HCC, most of the genetic alterations in TGFB1, GSTP1, and GSTM1 were amplifications, which may cause increased expression. However, the prognostic significance varied in this study. The amplification of TGFB1 indicated a worse OS in HCC, whereas amplifications of GSTM1 and GSTP1 (not statistically significant) indicated better prognosis. TGFB1 is a pleiotropic gene with a dual role in hepatocarcinogenesis: apoptosis induction in early phases, but promotion of tumorigenesis in cells with mechanisms to overcome the suppressor effects (Moreno-Caceres et al., 2017). Recent studies found that the expression of TGFB1 genes were downregulated in breast cancer cells treated with curcumin (Calaf & Roy, 2017). Glutathione S-transferases (GSTs) are a family of phase II detoxification enzymes that catalyze the conjugation of a wide variety of endogenous and exogenous toxins. A previous study showed that inter-individual GST variation plays a central role in reducing cell exposure to carcinogens (Di Pietro, Magno & Rios-Santos, 2010). For example, reduced GSTP1 expression may contribute to oxidative stress in HCC (Li et al., 2013). Similar results were observed in breast cancer cells and curcumin-activated GSTP1 expression via antioxidant response element (Nishinaka et al., 2007). However, studies linking curcumin and GST genes in HCC are rare, underscoring the need for further investigation in this field.

These results were partially confirmed by in vitro experiments. HepG2 and Hep3B were used as the experimental objects because of their advantages of fast growth and easy passage. The present study confirmed the inhibitory effect of curcumin on liver cancer cells, and this inhibition increased with time and concentration. In the liver HCC study (TCGA, Provisional), three genes (TGFB1, GSTP1, and GSTM1) showed amplification. For this reason, changes in these amplified genes were more easily detected using PCR. This was confirmed in the present study when it was found that two genes prone to amplification were suppressed in HCC cells treated with curcumin. The other genes, however, did not show changes in expression because the main means of mutation was not amplification mutations but truncating mutations and deep deletions. In terms of the relationship between curcumin and protein expression, only one protein change TGFB1 was detected by western blot. This may be because of the complexity of post-transcriptional regulation of mRNA, meaning that mRNA amplification may not be consistent with protein expression. The results of this study showed that TGFB1 expression was inconsistent at mRNA and protein levels after curcumin treatment, possibly because TGFB1 may be regulated by intracellular activators. In addition, although mRNA concentration is widely used as a surrogate for protein abundance, studies comparing mRNA and protein expression on a global scale indicate that mRNA levels only partly correlate with the corresponding protein concentrations. It has been estimated that protein concentrations are determined by the corresponding mRNA concentrations by only 20–40% (Nie, Wu & Zhang, 2006; Tian et al., 2004).

Furthermore, studies have proved that TGFB1 has a dual role in the development and progression of HCC. In the previous analysis, elevated mRNA levels of TGFB1 were correlated with poor prognosis in HCC patients. This may be because TGFB1 promoted the growth, invasion, and metastasis of HCC cells through the epithelial-mesenchymal transition (Bierie & Moses, 2006; Massagué, 2008; Matsuura et al., 2004). Regarding HCC cell levels, TGFB1 was a potent growth inhibitor and induces apoptosis in these cells, so it is regarded as a tumor-suppressive cytokine (Senturk et al., 2010).Therefore, further studies are needed to validate and establish the specific molecular mechanisms by which curcumin regulates HCC cells.

Conclusions

In summary, by using DrugBank, STRING, CDC, and cBioPortal databases, the connectivity between curcumin and HCC was discovered. Curcumin has the potential to become an alternative chemotherapy or chemoprevention treatment for HCC. The drug–target–cancer network analysis utilized in this study facilitated the testing and validation of reasonable hypotheses explaining curcumin-induced gene alterations in cancers by applying the available biological information in studies from bedside to bench. As advances in curcumin research using traditional experimental approaches continue, additional drug targets will undoubtedly be identified, leading to improved curcumin-related genetic networks, signaling pathways, and cancer types.

Supplemental Information

Supplemental Information 1 Supplemental Tables.

Table S1: Direct protein targets and indirect protein targets of curcumin. Table S2: PPI pairs in the network of curcumin-mediated proteins. Table S3: Node degrees of 204 curcumin-mediated proteins. Table S4: All Enrichment KEGG pathways. Table S5: 505 genes associated with HCC.

Click here for additional data file.

Supplemental Information 2 Raw data for the figures and analysis.

Click here for additional data file.

We would like to thank LetPub for its linguistic assistance during the preparation of this manuscript.

List of abbreviations

DPTs direct protein targets

IPTs indirect protein targets

PPI protein–protein interaction

KEGG Kyoto Encyclopedia of Genes and Genomes

CTD Comparative Toxicogenomics Database

HCC hepatocellular carcinoma

cBioPortal cBio Cancer Genomics Portal

OS overall survival

FAN functional/activity network

OD optical density

MTT (3-(4, 5)-dimethylthiahiazo (-z-y1)-3, 5-di-phenytetrazoliumromide)

Additional Information and Declarations

Competing Interests

Author Contributions

Data Availability

The authors declare that they have no competing interests.

Yuanyuan Zhao conceived and designed the experiments, performed the experiments, authored or reviewed drafts of the paper, and approved the final draft.

Jiahao Tao conceived and designed the experiments, performed the experiments, authored or reviewed drafts of the paper, and approved the final draft.

Zhuangzhong Chen conceived and designed the experiments, performed the experiments, prepared figures and/or tables, and approved the final draft.

Suihui Li analyzed the data, prepared figures and/or tables, and approved the final draft.

Zeyu Liu analyzed the data, prepared figures and/or tables, and approved the final draft.

Lizhu Lin conceived and designed the experiments, performed the experiments, authored or reviewed drafts of the paper, and approved the final draft.

Linzhu Zhai conceived and designed the experiments, performed the experiments, analyzed the data, authored or reviewed drafts of the paper, and approved the final draft.

The following information was supplied regarding data availability:

The data is available in the Supplemental Files.

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
