# Peer review of "Functional drug–target–disease network analysis of gene–phenotype connectivity for curcumin in hepatocellular carcinoma"

_PeerJ, doi:10.7717/peerj.12339_

## Round 0.1 · original submission · Major Revisions

Please consider all points raised by the referees when revising the manuscript.

Reviewer 1 ·

Basic reporting

The article titled “Functional drug–target–disease network analysis of gene–phenotype connectivity for curcumin in hepatocellular carcinoma” (#61606) by Zhao Y et al., emphasizes on identification of FAN analysis underlying the connectivity between curcumin and HCC using Drugbank, STRING, CDC, and cBioPortal databases and we discovered the 360 connectivity between curcumin and HCC. They have identified five genes as curcumin targets using this network analysis.

Experimental design

The Methodology and the study design justifies the rationale of this work.

Validity of the findings

Overall, the findings were of interest to the peers working in HCC and other related cancers but there were several areas that need authors attention to detail.

Additional comments

As a reviewer from a reader’s point of view the below comments were made to improve the overall reach of this work.
Major concerns:
1. The introduction is very short, and it is not adequately providing the background of the disease and the significance of the work. The authors may think of providing more information on several aspects relevant to this work like about HCC itself, statistics of this disease (# of deaths), which age groups are affected, gender differences if exist, factors leading to or effecting the disease, current interventions etc.
2. Line 271: The RNA expression levels are highly significant in Figure 7C, but this did not translate to protein levels Figure 7D and 7E. Although the authors discussed a bit (Line 350) about this in discussion it is not satisfactory as the differences are huge.
3. Line 271: Why did the authors exclude GSTP1 in checking protein expression levels like they have done for TGFb? In the RTqPCR data (Figure 7C) GSTP1 is also significantly low like TGFb? This need to be verified at protein levels.
4. Line 271: Figure 7: The data that was shown here is for 20 µM concentration of Curcumin but in the original data/raw data in Supplementary Figures S2 they have shown data of 30 µM concentration. This is misleading and confusing and raises doubt on the quality of the data and results presented.
5. Figure 7 and Supplementary Figure S2: The raw western blot image shown in Figure S2 and the Western blot image shown in Figure 7D are very different. This is a serious concern regarding the trustworthiness of the data presented.
6. Figure 7A: The authors have seen a significant reduction in cell viability at 30 µM concentration and as early as 24 h in both Hep3B and HepG2 cells. In that case, it would be ideal to use these conditions for the further RTqPCR and protein expression levels. Testing at these conditions would provide an accurate answer for the differences they have seen at RNA and protein levels of TGFb.
7. Figure 1B: What is the reason that all the other 4 genes are connected to GSTP1 and it is down regulated in Figure 7C? It would be interesting to investigate this gene at protein level
8. Suggestion: It would be very helpful for the reader if the authors can provide a workflow of the different analysis they have performed as a flow chart with identification of genes at each step
Minor:
1. Line 48: Correct “was” to “were”
2. Line 53: Elaborate “MTT” in the first appearance in the manuscript text
3. Line 60: Elaborate “OS” in the first appearance in the manuscript text
4. Line 59: Replace special characters in the parentheses to commas
5. Line 125: Elaborate FDR in the first appearance in the manuscript text
6. Line 150: Cell Cook (Provide company details in parentheses)
7. Line 155: Elaborate MTT in the first appearance in the manuscript text
8. Line 159: CCK-8, what is this solution, provide details and elaborate CCK
9. Line 160: Delete space between 4 h.
10. Line 161: Delete space between 450 nm
11. Line 173: Change “an” to “a”
12. Line 177: Write ingredients in RIPA buffer
13. Line 181 and 182: Delete space before and after parentheses
14. Line 184: Which HRP-IgG secondary antibody is used from which company?
15. Line 186: write company name of chemiluminescence system that was used?
16. Line 195: Provide references for each activity like anti-tumor, antibacterial etc
17. Line 196: Provide clinical trial number here.
18. Line 234: Based on the Figure 3A the authors have identified 504 target genes by CTD but in the text they have mentioned 505. Correct the same.
19. Line 244: The alterations of gene sets from 0-29% were not clear from the figure 4
20. Line 254: Delete comma after GSTM1
21. Line 258: 364 patients were involved. Please mention about the patient details in the materials and methods sections and include relevant approvals. Is this a clinical study performed by the authors?
22. Line 265: The authors has mentioned that “For Hep3B cells, curcumin (10 μM) can significantly decrease cell viability after 24, 48, and 72 h” but in the data Figure 7A it is not significant for 24 h time point. Correct the same.
23. Line 272: Change Rb1 to RB1
24. Line 282: Change the font to italics to in vitro and in vivo
25. Figures S1 and S2 and Tables S5-9 were not cited in the manuscript text. Cite them at relevant locations
26. Line 315: Change the font to italics to in vitro
27. Line 330: The font and size of the citation (Moreno-Caceres et al., 2017) need to be changed to match the test of the text
28. Line 331: Change “Was” to “Were”
29. Line 359: Correct the spelling of DrugBank
30. Table 1: The gene functions are not correctly annotated to the genes. Please verify and correct
31. Table 2: Either write Gene or Gene name. Maintain uniformity
32. Table 3: Last row: add space between “and atherosclerosis”
33. Figure 1B: There is only one type of blue line. No other type blue line is seen. Please change to contrasting colors rather than similar ones.

Reviewer 2 ·

Basic reporting

The article is well written and it has a good flow throughout. The article is easy to comprehend, with all the sections (abstract, introduction, experimental data and inferences) carefully planned and clearly described. The figures, raw data and supplementary material are in line with the journal standards. The citations are current and relevant to the study. One suggestion that the authors could look into addressing would be in terms of reducing the words in the discussion section. It would be helpful if the discussion were made more succinct with less repetition from what has been described in the introduction/results sections of the article.

Experimental design

I commend the authors for the careful design and well-executed experiments, both in terms of the bioinformatics data as well as the in vitro experiments. The in vitro cell viability experiments lend proof of concept of the hypothesis and conclusions from the FAN analysis.
The study is supported by thorough analysis, explanation of the techniques used, their relevance and the inferences are sound. The FAN analysis described will be useful in further studies addressing effects of curcumin or other therapeutic agents in the context of multiple disease states.

Validity of the findings

The rationale for the inferences from the experimental data is very clear and practical. The inclusion of the Kaplan-Meier analysis and the in vitro experiments (cell viability and gene expression) certainly corroborate to strengthen the findings and pave way for future studies of effects curcumin in cancer. One point I would like to raise with regard to the in vitro experiments are the protein expression levels of TGFB1. The gel images are not very convincing of the fact that curcumin has an effect on this particular protein expression, although curcumin does appear to have an effect on cell viability. The question I would like for the authors to address is if the protein expression was carried out at 72h timepoint wherein there is a significant decrease in cell viability. If the gel image is a representation of the experiments at 72h timepoint, I would suggest that the authors refrain from commenting/inferring much on the levels of protein expression modulated by curcumin. As the authors have rightly suggested, it may also be modulated due to factors/pathways independent of the curcumin related pathways.
The supplementary materials are very informative and provide sound justifications for the authors’ choice of following up on cancer-related pathways and HCC to dissect the therapeutic potentials of curcumin.

Additional comments

The study is relevant in terms of the experiments and the inferences made. The findings of the study do indeed provide valuable insights into the potential anti-tumorigenic properties of curcumin, especially in the context of HCC.
The study results are promising in terms of potential future studies related to the therapeutic effects of curcumin not only in HCC but also in other cancers, in lieu of the list that the authors have reported (supplementary materials) pertaining to cancer-related genes and biomarkers.

·

Basic reporting

In the manuscript (#61606) entitled, “Functional drug–target–disease network analysis of gene–phenotype connectivity for curcumin in hepatocellular carcinoma”, the authors have have aimed to explore the underlying mechanisms of curcumin and broaden the perspective of targeted therapies. They have used an extensive data-driven approach to support their experimental findings.

Experimental design

The reporting of the methods in the manuscript is robust and the overall manuscript and study design is comprehensive and well thought through. The data presented/experiments performed are rigorous enough and does justify the conclusions.

Validity of the findings

The authors have performed rigorous scientific analyses with the help of multiple online modalities and databases and have performed various validation assays. The experimental findings are conclusive and very well supported by data. Enough experimental evidences have been given to support the results.

Additional comments

The authors have used a well-planned and comprehensive approach to come up with a logical conclusion and the manuscript is well written and can be published in its present form in this journal with some minor changes as detailed below.
The authors need to address the issue just one main issue about this manuscript and it is the language used. I could find many grammatical errors, typographical errors, punctuation mistakes, confusing sentences and a peculiar choice of words in many instances making the comprehension of data difficult. Thus the manuscript needs to be edited and prrof-read thoroughly.
Introduction needs citations to new articles especially with regard to the chemopreventive effects of curcumin.
Line 183 The samples were washed is incorrect, it should be membrane was washed.

---

## Round 0.2 · Minor Revisions

The authors have adequately replied to the issues raised by the referees. Please still consider the point made by one referee.

The authors state in the introduction “However, the precise molecular mechanisms underlying the anti-tumor and chemopreventive activities of curcumin have yet to be elucidated.” This is only partially true since several molecular mechanisms have been described in much detail. The authors must cite work showing the anti-NFkB activity of curcumin as well as its inhibitory action on the proteasomal kinase DYRK2 and ATP-synthase.

·

Basic reporting

In the manuscript (#61606) entitled, “Functional drug–target–disease network analysis of gene–phenotype connectivity for curcumin in hepatocellular carcinoma”, the authors have have aimed to explore the underlying mechanisms of curcumin and broaden the perspective of targeted therapies. They have used an extensive data-driven approach to support their experimental findings.

Experimental design

The reporting of the methods in the manuscript is robust and the overall manuscript and study design is comprehensive and well thought through. The data presented/experiments performed are rigorous enough and does justify the conclusions.

Validity of the findings

The authors have performed rigorous scientific analyses with the help of multiple online modalities and databases and have performed various validation assays. The experimental findings are conclusive and very well supported by data. Enough experimental evidences have been given to support the results.

Additional comments

The changes made as per the peer review have made the manuscript better and more understandable to the reader. Although, Line 200 The samples were washed is still incorrect, it should be membrane was washed. Please correct.

---

## Round 0.3 · accepted · Accept

I appreciate that you have adequately addressed all issues raised.